# GPR30 Alleviates Pressure Overload-Induced Myocardial Hypertrophy in Ovariectomized Mice by Regulating Autophagy

**DOI:** 10.3390/ijms24020904

**Published:** 2023-01-04

**Authors:** Shuaishuai Zhang, Jipeng Ma, Xiaowu Wang, Diancai Zhao, Jinglong Zhang, Liqing Jiang, Weixun Duan, Xiaoya Wang, Ziwei Hong, Zilin Li, Jincheng Liu

**Affiliations:** Department of Cardiovascular Surgery, Xijing Hospital, Fourth Military Medical University, Xi’an 710032, China

**Keywords:** GPR30, pressure overload, myocardial hypertrophy, ovariectomy, autophagy

## Abstract

The incidence of heart failure mainly resulting from cardiac hypertrophy and fibrosis increases sharply in post-menopausal women compared with men at the same age, which indicates a cardioprotective role of estrogen. Previous studies in our group have shown that the novel estrogen receptor G Protein Coupled Receptor 30 (GPR30) could attenuate myocardial fibrosis caused by ischemic heart disease. However, the role of GPR30 in myocardial hypertrophy in ovariectomized mice has not been investigated yet. In this study, female mice with bilateral ovariectomy or sham surgery underwent transverse aortic constriction (TAC) surgery. After 8 weeks, mice in the OVX + TAC group exhibited more severe myocardial hypertrophy and fibrosis than mice in the TAC group. G1, the specific agonist of GPR30, could attenuate myocardial hypertrophy and fibrosis of mice in the OVX + TAC group. Furthermore, the expression of LC3II was significantly higher in the OVX + TAC group than in the OVX + TAC + G1 group, which indicates that autophagy might play an important role in this process. An in vitro study showed that G1 alleviated AngiotensionII (AngII)-induced hypertrophy and reduced the autophagy level of H9c2 cells, as revealed by LC3II expression and tandem mRFP-GFP-LC3 fluorescence analysis. Additionally, Western blot results showed that the AKT/mTOR pathway was inhibited in the AngII group, whereas it was restored in the AngII + G1 group. To further verify the mechanism, PI3K inhibitor LY294002 or autophagy activator rapamycin was added in the AngII + G1 group, and the antihypertrophy effect of G1 on H9c2 cells was blocked by LY294002 or rapamycin. In summary, our results demonstrate that G1 can attenuate cardiac hypertrophy and fibrosis and improve the cardiac function of mice in the OVX + TAC group through AKT/mTOR mediated inhibition of autophagy. Thus, this study demonstrates a potential option for the drug treatment of pressure overload-induced cardiac hypertrophy in postmenopausal women.

## 1. Introduction

With the global trend in aging, the incidence of heart failure (HF) caused by hypertension and senile degenerative cardiac valvular disease has increased significantly [1,2]. It should be noted that there are significant gender differences in heart failure: the incidence of HF in postmenopausal women is markedly increased than in men of the same age [3,4]. As many studies have shown that HF is closely related to myocardial hypertrophy and fibrosis [5,6,7,8], estrogen and its receptors might play a vital role in antihypertrophy and antifibrosis of the heart. However, studies on the effect of estrogen replacement therapy on the cardiovascular system often draw contradictory conclusions [9,10,11,12]. Therefore, the mechanism of postmenopausal myocardial hypertrophy, the role of estrogen and its receptors needs further elucidation.

G Protein Coupled Receptor 30 (GPR30), a seven-transmembrane novel estrogen receptor, is located on both the cell membrane and endoplasmic reticulum, and is widely distributed in many tissues including the heart, lung, ovary, brain and so on [13,14,15]. G1, the specific agonist of GPR30 [16], was reported to improve contractile function and reduced infarct size in mice hearts subjected to ischaemia/reperfusion injury [17]. Previous studies in our group have shown that GPR30 can also attenuate myocardial fibrosis and reduce the myocardial infarct area in ovariectomized (OVX) mice [18]. Recently, we found that GPR30 reduces transverse aortic constriction (TAC)-induced myocardial fibrosis in aged female mice by inhibiting the ERK1/2-MMP-9 signaling pathway [19]. However, the effect of GPR30 on TAC-induced myocardial hypertrophy in OVX mice and the underlying mechanism have not been investigated yet.

Autophagy is an evolutionarily conserved process in which cytoplasmic proteins and damaged organelles are sequestered in an autophagosome and become degraded after fusion with the lysosome [20,21]. Many studies reveal that autophagy is closely related to heart diseases [22,23], including cardiac hypertrophy [24,25]. Autophagy has been found to be an obligatory element in pathological cardiac remodeling, and HDAC inhibitors attenuate cardiac hypertrophy by suppressing autophagy [26]. Constitutive autophagy is considered a homeostatic mechanism for maintaining normal cardiac structure and function, and under hemodynamic stress changes, autophagy is enhanced to protect the heart from pathological remodeling [27,28]. Despite these previous studies, the relationship between autophagy and GPR30/G1 in TAC-induced cardiac hypertrophy in OVX mice is still not clear.

In this study, we use TAC surgery to establish the model of Pressure Overload (PO)-induced myocardial hypertrophy and we investigate the effect of GPR30 on pathological myocardial hypertrophy in OVX mice and the mechanism behind it.

## 2. Results

### 2.1. Effects of Pressure Overload on Cardiac Hypertrophy and Fibrosis in Sham and OVX Mice

To study the effect of pressure overload on cardiac hypertrophy and fibrosis in OVX mice, bilateral ovariectomy was carried out, and TAC surgery was performed two weeks later. At 8 weeks after TAC, HE, wheat germ agglutinin (WGA) and Masson staining were used to compare the effect of TAC on myocardial remodeling among different groups. HE staining showed that the ventricular wall thickness and the size of the heart were increased in the OVX + TAC group compared with the Sham, OVX and TAC groups (Figure 1A). Masson staining showed that myocardial fibrosis was more severe in mice of the TAC group than in the Sham group and was more severe in the OVX + TAC group than in the TAC group (Figure 1A,B). The WGA staining results show that the mean cross-sectional area of cardiomyocytes was increased in the TAC group compared with the CON group and was increased in the OVX + TAC group compared with the TAC group (Figure 1A,C).

### 2.2. Effects of Pressure Overload on Cardiac Function, HW/BW and LW/BW after 8 Weeks of TAC

M-mode echocardiography was used to assess cardiac function and measure ventricular wall thickness. As shown in the echocardiographic results, compared with the TAC groups, the ejection fractions (EF%) and fraction shortening (FS%) were both decreased in the OVX + TAC group (Figure 2A–C), while interventricular septal thickness at end diastole (IVSd) and left ventricular posterior wall thickness at end diastole (LVPWd) were both significantly increased in the OVX + TAC group compared with the TAC group (Figure 2D,E). Furthermore, the heart weight to body weight ratio (HW/BW) and lung weight to body weight ratio (LW/BW) in the OVX + TAC group were both significantly increased compared with the TAC group (Figure 2F,G), which was consistent with the results of HE staining and WGA staining (Figure 1A,C). These results combined with the results of Figure 1 imply that TAC in ovariectomized mice could exacerbate pathological cardiac hypertrophy and myocardial fibrosis compared with TAC in sham mice, and the OVX + TAC mice showed more serious cardiac function decline than the OVX mice or TAC mice. Similarly, cardiac fibrosis was worsened in the OVX + TAC group compared with the TAC group, indicating that estrogen and its receptors might play a protective role in TAC-induced cardiac hypertrophy and fibrosis.

### 2.3. GPR30 Agonist G1 Alleviated Cardiac Hypertrophy and Fibrosis of Mice in the OVX + TAC Group

As a novel estrogen receptor, GPR30 has been demonstrated to have protective effects on the cardiovascular system. To study the effect of GPR30 and its specific agonist G1 on cardiac hypertrophy and myocardial fibrosis in the OVX + TAC mice, G1 was injected intraperitoneally every two days with a dose of 35 μg/kg. After 8 weeks of G1 treatment, cardiac hypertrophy was alleviated in the OVX + TAC + G1 group compared with the OVX + TAC group, as evidenced by decreased ventricular wall thickness in the HE staining (Figure 3A), less fibrosis area in the Masson staining (Figure 3A,B) and smaller size of myocardial cells in the WGA staining (Figure 3A,C). The histological results indicate that G1 treatment attenuated myocardial hypertrophy and fibrosis in the OVX + TAC mice. It should be noted that no significant difference on cardiac hypertrophy and fibrosis was found between the OVX + G1 and OVX groups in our study (Figure 3A–C).

### 2.4. G1 Treatment Improved Cardiac Function of OVX + TAC Mice

Consistent with the histological results, the echocardiographic data show that EF and FS were both higher in the OVX + TAC + G1 group than in the OVX + TAC group (Figure 4A–C). IVSd and LVPWd were both decreased in the OVX + TAC + G1 group compared with the OVX + TAC group (Figure 4D,E). HW/BW and LW/BW of the OVX + TAC + G1 mice were significantly decreased compared with the OVX + TAC mice (Figure 4F,G). Therefore, the histological and echocardiographic results both indicate that G1 treatment could, at least partially, recover cardiac function and attenuate myocardial hypertrophy and fibrosis in the OVX + TAC mice, showing an effect of cardioprotection. In addition, there was no significant difference in cardiac function (Figure 4A–C), HW/BW (Figure 4F) or LW/BW (Figure 4G) between the OVX + G1 and OVX groups in our study.

### 2.5. G1 Decreased the Autophagy Level of Myocardial Tissue in OVX + TAC Mice

Many studies have confirmed that autophagy plays an important role in cardiac function and metabolism [24,25], and G1 shows a protective effect by regulating autophagy intensity in the nervous system [29,30]. In order to verify whether the cardiac protection of the G1 treatment in our study was related to autophagy, we detected the expression of LC3 and P62, which are reported as markers of autophagy [14]. The Western blot results revealed that compared with Sham group, LC3II expression was significantly enhanced in the TAC group and LC3II expression in the OVX + TAC group was significantly higher than that in the TAC group (Figure 5A,B). In addition, the expression of LC3II demonstrated a statistically significant decrease in the OVX + TAC + G1 group compared with that in the OVX + TAC group, and P62 expression was markedly increased in the OVX + TAC + G1 group compared with the OVX + TAC group (Figure 5A–C). Additionally, there was no significant difference in the expression of LC3II and P62 among the Sham, OVX and OVX + G1 groups (Figure 5A–C). These results indicate that the autophagy level in the OVX + TAC group was increased compared with the TAC group and G1 could reduce the autophagy level in OVX + TAC mice.

### 2.6. G1 Attenuated AngII-Induced Hypertrophy of H9c2 Cells In Vitro

To further explore the mechanism of G1 treatment alleviating myocardial hypertrophy in the OVX + TAC group, we conducted experiments in vitro to clarify the role of autophagy and the molecular mechanism mediating this process. AngII was used to induce H9c2 hypertrophy. F-actin of H9c2 cells were stained with FITC-phalloidin and examined by fluorescence microscopy. Consistent with the results in vivo, AngII could induce cardiomyocyte hypertrophy and increase the expression of hypertrophic genes (Appendix A) [31,32,33], and G1 attenuated H9c2 cells hypertrophy and downregulated the expression of hypertrophy genes (Appendix A and Figure 6A,B). It should be noted that G1 did not change the cell area of H9c2 without AngII since there was no significant difference between the G1 and CON groups in cell area or hypertrophic gene expression (Appendix A and Figure 6A,B).

### 2.7. The Akt/mTOR Pathway and Autophagy Were Involved in G1 Alleviating AngII-Induced Hypertrophy of H9c2 Cells

To further explore the mechanism of G1 protecting cardiac hypertrophy, proteins of autophagy and related pathways were analyzed. The Western blot results show that the p-AKT/AKT and p-mTOR/mTOR ratios were significantly reduced in the AngII group compared with the Con group and were remarkably increased in the AngII + G1 group compared with the AngII group (Figure 7A–C). Moreover, AngII could promote the autophagy level of H9c2 cells, whereas G1 treatment decreased the autophagy of H9c2 in the AngII group (Figure 7A,D,E), which was similar with the results in vivo. As the Akt/mTOR pathway was reported to be involved in autophagy modulation and GPR30 activation was reported to be associated with AKT phosphorylation, we supposed that autophagy might be regulated through this pathway in this study. Our results show that the phosphorylation of both AKT and mTOR in the AngII group were significantly decreased compared with the AngII + G1 and Con groups (Figure 7B,C). This result suggests that the Akt/mTOR pathway might mediate the effect of G1 on autophagy regulation.

### 2.8. G1 Decreased the Autophagic Flux of H9c2 Cells Enhanced by AngII

To assess the autophagy flux changes in H9c2 cells in these groups, we conducted mRFP-GFP-LC3 confocal fluorescence microscopy and puncta analysis. In Figure 8A, the yellow fluorescent puncta represent autophagosomes and the red fluorescent puncta represent autolysosomes in the merged pictures, indicating early and late stages of autophagy, respectively. Our statistical analysis from the merged pictures shows that the AngII treatment significantly increased the yellow and free red fluorescent puncta compared with control group, indicating enhanced autophagic flux (Figure 8A,B). Additionally, there were significantly fewer yellow fluorescent puncta and free red fluorescent puncta in the AngII + G1 group than in the AngII group (Figure 8A,B), which means G1 decreased the AngII-enhanced autophagic flux. Therefore, autophagy might be an important mechanism of G1 protecting H9c2 from hypertrophy in vitro. As G1 could decrease the number of autophagosomes and autolysosomes of H9c2 cells in the AngII group at the same time (Figure 8A,B), we infer that G1 might work on the initial stage of autophagy to inhibit the autophagic flux in vitro, which is consistent with the result that the AKT/mTOR pathway might mediate the autophagy changes in the AngII and AngII + G1 groups (Figure 7). To sum up, these results show that the AngII treatment could induce H9c2 hypertrophy and increase the autophagic flux, while G1 treatment could restore the autophagic flux and at least partly reverse hypertrophy of H9c2 cells, which is regulated by the AKT/mTOR pathway.

### 2.9. LY294002 or Rapamycin Blocked the Effect of G1 Decreasing the Elevated Autophagy Induced by AngII in H9c2 Cells

To further confirm the relationship of the AKT/mTOR pathway and autophagy in G1 protection of H9c2 cells hypertrophy, the PI3K inhibitor LY294002 and mTOR inhibitor rapamycin were used to inhibit the activation of the AKT/mTOR pathway. After LY294002 or rapamycin treatment, the phosphorylation of AKT and mTOR of cells in the AngII + G1 + LY294002 group was significantly downregulated compared with cells in the AngII + G1 group (Figure 9A–C), while LC3II expression was markedly increased and P62 expression was significantly decreased in H9c2 cells from the AngII + G1 + LY294002 or rapamycin group compared with cells in the AngII + G1 group (Figure 9A,D,E). With these results, we conclude that the AKT/mTOR pathway mediated the regulation of autophagy in this study.

### 2.10. Effect of LY294002 or Rapamycin on Cardiac Hypertrophy of H9c2 Cells in AngII + G1 Group

To verify the relationship between cell autophagy and cardiomyocyte hypertrophy in this research, the mean cell area and cardiac hypertrophy marker genes (Appendix A) were analyzed and compared among the different groups after adding LY294002 or rapamycin in the AngII + G1 group. The results show that both LY294002 and rapamycin could abolish the effect of G1 alleviating H9c2 cells hypertrophy induced by AngII, as shown in Figure 10A,B. Collectively, these findings, combined with the results in Figure 9, suggest that G1 might alleviate AngII-induced cardiomyocyte hypertrophy in vitro by inhibiting autophagy through activating the AKT/mTOR pathway.

## 3. Discussion

In this study, we investigated the role of GPR30 in the pathological cardiac hypertrophy of OVX mice, and our results include several important findings regarding the effect of GPR30 on myocardial hypertrophy and the mechanism behind it: (1) We found that cardiac hypertrophy was more significant in the OVX + TAC group than in the OVX and TAC groups. (2) GPR30 activation by G1 could alleviate cardiac hypertrophy of the OVX + TAC mice. (3) Autophagy was significantly increased in the OVX + TAC group compared with the OVX and TAC groups. (4) G1 decreased the autophagy of myocardial tissue in the OVX + TAC group and reversed cardiac hypertrophy. (5) Our in vitro results show that changes in the autophagy of H9c2 cells were at least partly mediated by AKT/mTOR signaling pathways. (6) The AKT/mTOR pathways were also involved in G1 attenuating myocardial hypertrophy in OVX + TAC mice in vivo (Appendix A). Collectively, these results show that GPR30 activation protected TAC induced cardiac hypertrophy in the OVX mice by regulating autophagy through the AKT/mTOR pathways.

Both physiological and pathological stimuli could result in cardiac hypertrophy [34]. Regular exercise often leads to physiological cardiac growth, which appears to mediate cardiac benefits. In contrast, diseases such as hypertension and aortic stenosis lead to pathological cardiac hypertrophy, which often precedes heart failure and is associated with adverse outcomes [34]. Pathological myocardial hypertrophy is often closely related to myocardial fibrosis, cardiomyocyte apoptosis and decline of cardiac function, which was also observed in our research [35]. In our study, TAC not only led to cardiac hypertrophy but also cardiac fibrosis. Moreover, a loss of estrogen by ovariectomy further aggravated myocardial hypertrophy and myocardial fibrosis, which could be partly reversed by activation of GPR30. Bhuiyan MS et al. reported that the LW/BW ratio was significantly increased in the OVX + TAC group compared with OVX and TAC groups [36], which is consistent with our results.

Wang et al. found that G1 attenuated OVX-induced LV hypertrophy and improved the diastolic function in vivo in rats and G1 prevented angiotensin II-induced hypertrophy in H9c2 cardiomyocytes in vitro [17]. It should be noted that OVX alone could induce LV hypertrophy in their study, whereas in our study, there was no statistical difference between the sham group and OVX group in terms of cardiac hypertrophy. Two reasons might explain this difference. Firstly, there might be differences in the cardiovascular responsiveness to a loss of estrogen between rats and mice. Secondly, in Wang’s study, rats were randomly assigned to undergo either OVX or a sham operation at 4 weeks of age, while mice in our research undergoing OVX were at the age of 8 to 10 weeks.

Autophagy is a highly conserved physiological process, which has been proved to be closely related to myocardial hypertrophy and fibrosis [27,28,34]. Although autophagy has been reported to play important roles in the maintenance of homeostasis in the heart, the relationship between autophagy and myocardial hypertrophy is not very clear. In this study, we found that autophagy was involved in the protective effect of GPR30 on myocardial hypertrophy. To our best knowledge, this is the first study showing that autophagy is involved in the mechanism of GPR30 protecting PO-induced cardiac hypertrophy in OVX mice in vivo. It was reported that stresses such as hunger, reactive oxygen species (ROS) or misfolded proteins, could enhance the intensity of myocardial autophagy [37]. However, suppression of autophagy is commonly observed in aging hearts with increased oxidative stress, misfolded proteins, and dysfunctional mitochondria [38,39,40]. One possible explanation for this paradox is that continuous activation of autophagy caused by higher oxidative stress and protein misfolding might lead to exhaustion of the autophagic machinery, eventually causing suppression of autophagy [27]. In our study, after ovariectomy, without the protection of estrogen, consistent pressure overload led to more severe myocardial hypertrophy and increased autophagy in the OVX + TAC mice compared with the TAC mice. It seems that autophagy in myocardial tissue had not reached the stage of exhaustion in the TAC or OVX + TAC groups after 8 weeks. In fact, some studies have shown that 8 weeks was the usual duration of myocardial hypertrophy induced by TAC when heart failure and severe cardiac dysfunction had not yet occurred [41,42]. Therefore, the enhancement of autophagy might be a maladaptive change in the face of the dual stress stimulation of OVX and TAC [43].

It has been proved that G1 activation plays an important role in protecting the cardiovascular system, glucose and lipid metabolism, neuroprotection and immunity [44]. Geetanjali et al. found that G1 treatment of ovariectomized female mice reduced body weight and improved glucose homeostasis without changes in food intake, fuel source usage, or locomotor activity. G1 treated female mice also exhibited increased energy expenditure, lower body fat content, and reduced fasting cholesterol, glucose, insulin, and inflammatory markers [16]. Yue et al. demonstrated that GPR30 activation reduced glutamate-induced excessive autophagy in neurons and protected neurons against excitotoxicity [29]. Wang et al. reported that activation of GPR30 by G1 played a neuroprotection role via the regulation of astrocyte autophagy and supported astrocytic GPR30 as a potential drug target against ischemic brain damage [30]. Brunsing et al. showed that G1 could elicit IL-10 expression in T helper type 17 cells and exerted an anti-inflammatory effect [45]. Our results show that the activation of GPR30 has an important protective effect on PO-induced myocardial hypertrophy and fibrosis in OVX mice, which provides an important potential target for myocardial protection in menopausal women. Unlike traditional estrogen replacement therapy, which activates the classical estrogen receptors Erα and ERβ, it will not produce corresponding side effects and has the potential to become a more precise drug treatment.

There are a multitude of downstream effectors following the activation of GPR30 within the cell, such as ERK1/2 with the involvement of transactivation of the EGFR [46] or cyclic adenosine monophosphate with activation of adenylyl cyclase [47] and the PI3K/Akt pathway [48]. In addition, GPR30 could activate eNOS to release NO [49], thus playing a role in vasodilation [50]. Among these downstream signal pathways, the PI3K/Akt pathway was reported to be related to autophagy regulation [51]. Similarly, the Akt/mTOR pathway was detected in our research when autophagy was involved in the mechanism of G1 protecting cardiac hypertrophy. Our in vitro results show that the Akt/mTOR pathway mediated the regulation of autophagy when GPR30 was activated.

It should be noted that although we successfully observed changes in myocardial hypertrophy and autophagy 8 weeks after TAC in this experiment, multiple observation time points might show dynamic changes in myocardial hypertrophy and autophagy. In addition, as a cardioprotective drug that might be used clinically, the toxicity and adverse reactions of G1 on other organs such as the liver, kidney and lung should be thoroughly studied in future research.

In conclusion, our present research showed that GPR30 activation by G1 could attenuate TAC-induced cardiac hypertrophy and fibrosis in OVX mice through regulating the autophagy of myocardial tissue by the Akt/mTOR pathway. Our research reveals that GPR30 has a protective role on the myocardial hypertrophy and fibrosis of OVX mice, which has potential as a drug target for the treatment of postmenopausal women. Further studies regarding the effects and mechanisms of GPR30 activation on the cardiovascular system are needed to accelerate the clinical transformation of G1.

## 4. Materials and Methods

### 4.1. Reagents

GPR30 agonist G1 (10,008,933) was purchased from Cayman Chemical (Ann Arbor, MI, USA). GAPDH was obtained from Santa Cruz Biotechnology (Santa Cruz, CA, USA). The rabbit anti-goat, goat anti-rabbit, and goat anti-mouse secondary antibodies were purchased from Zhongshan Company (Beijing, China). DAPI was obtained from Roche Molecular Biochemicals (Bayer, Mannheim, Germany). The ECL reagent was purchased from Millipore (Billerica, MA, USA). Anti-LC3B antibody(ab192890), F-actin(ab205), anti-BCL-2 antibody (ab182858), anti-BAX antibody(ab32503), and abti-P62 antibody (ab109012) were purchased from Abcam (Cambridge, MA, USA); AKT(9272), p-AKT(4060), mTOR(2983) and p-mTOR(5536) were purchased from Cell Signaling Technology (Beverly, MA, USA); AngII, Ad-mCherry-GFP-LC3 and rapamycin were purchased from Hanheng Biotechnology Co., Ltd. (Shanghai, China).

### 4.2. Experimental Animals

Female C57BL/6 mice (18–22 g, 8 to 10 weeks old, obtained from the Experimental Animal Center of Fourth Military Medical University, Xi’an, China) were housed in individual cages (3 to 5 mice per cage) under a 12:12-h light/dark cycle (light on 06:00) at 22–24 °C and had access to a regular pellet diet ad libitum. All mice used in this study were raised and used in accordance with the Guide for the Care and Use of Laboratory Animals of the Chinese Animal Welfare Committee and with the approval and supervision from the Fourth Military Medical University Committee on Animal Care.

### 4.3. Ovariectomy and Transverse Aortic Constriction(TAC) Surgery

The bilateral ovariectomy was performed on C57BL/6 female mice after being anesthetized in an induction chamber with 2% isoflurane mixed with pure oxygen (0.5–1.0 L/min). The TAC surgery was performed to establish the mouse model of chronic cardiac hypertrophy as previously described [52]. Briefly, mice were anesthetized in an induction chamber with 2% isoflurane mixed with pure oxygen (0.5–1.0 L/min). The animals were intubated endotracheally and connected to a ventilator (Minivent Type 845, Hugo Sachs Electronic, March, Germany, 100–120/min 0.15-mL tidal volume). A median thoracotomy was performed at the second intercostal space. The transverse aorta was constricted using a 7–0 silk suture ligature tied firmly against a 27-gauge needle between the carotid arteries. The needle was then removed immediately. In the sham-operated mice, the same surgical procedure was performed except for the ligation of the aorta. The chest was closed with a 6–0 silk suture, and a 5–0 silk suture was used to close the skin. After the surgery, the mice were kept warm on a 38 °C constant temperature blanket and carefully monitored until good recovery.

### 4.4. Experimental Design and Treatment

In order to investigate the effect of PO on myocardial hypertrophy and fibrosis in OVX mice, mice of 8 to 10 weeks were randomly divided into the following four experimental groups: sham group, OVX group, TAC group, OVX + TAC group. TAC surgery was performed in mice 2 weeks after the bilateral ovariectomy (OVX + TAC group). The second experiment was designed to study the effects of the specific agonist of GPR30, G1, on the myocardial hypertrophy and fibrosis of ovariectomized mice. Mice were randomly allocated to four groups: OVX, OVX + TAC, OVX + G1, and OVX + TAC + G1. After TAC surgery, G1 (35 μg/kg/d) was intraperitoneally injected for 8 weeks in the OVX + TAC + G1 and OVX + G1 groups, while mice in the other groups were given the same amount of vehicle (DMSO:PBS(V/V) = 1:99) treatment. After 8 weeks, the mice in each group were anesthetized with 1–2% isoflurane and their hearts were harvested for HE staining, Masson staining, WGA staining, and Western blot analysis.

### 4.5. H9c2 Cells Culture and In Vitro Study Design 

The rat myocardial cell line H9c2 [53] was obtained from the Cardiovascular Surgery Laboratory of the First Affiliated Hospital of the Fourth Military Medical University. H9c2 cells were cultured in Dulbecco’s Modified Eagle’s Medium High glucose (DMEM) (Sigma–Aldrich Co. LLC, St. Louis, MO, USA) supplemented with 10% fetal bovine serum (FBS) (Thermo Fisher Scientific Inc., Waltham, MA, USA) with penicillin and streptomycin. Ang II, G1(10 nM), rapamycin, and LY294002 were dissolved in dimethyl sulfoxide (DMSO, Sigma-Aldrich). The H9c2 cells were serum-starved for 12 h before applying the stimuli. For the first part of the cell experiment, H9c2 cells were divided into four groups: The cells in the control group (CON group) were treated with vehicle. The cells in the AngII group were treated with 0.1 μM AngII for 24 h to induce cell hypertrophy, and the cells in the AngII + G1 group were firstly treated with 10 nM G1 for 24 h and then incubated with 0.1 μM AngII for 24 h. The cells in the G1 group were treated with 10 nM G1 for 24 h only. After that, cells were collected for Western blot and immunofluorescence analysis. For tandem mRFP-GFP-LC3 fluorescence analysis, the cells were transfected with adenovirus before AngII or G1 treatment, as described below in Section 4.10. For the second part of the cell experiment, LY294002 (5 µM) or rapamycin (20 nM) was added 2 h before the AngII or G1 treatment; the following procedures were the same as in the first part of the experiment.

### 4.6. Echocardiographic Assessment of Cardiac Function

The echocardiographer was not informed of the details of experiment. After the mice were anesthetized with 1–2% isoflurane, mouse cardiac function was evaluated by transthoracic ultrasonography using a VisualSonics 2100 echocardiograph (FUJIFILM VisualSonics, Toronto, ON, Canada). Images of the parasternal long axis and short axis of the left ventricle were obtained by a 30-MHz transducer. The ejection fraction (EF)% and fraction shortening (FS)% (indicating the cardiac function), interventricular septal thickness at end diastole (IVSd), and left ventricular posterior wall thickness at end diastole (LVPWd) (indicating the thickness of ventricular wall) were detected and recorded. Then the above parameters were calculated by Vevo Lab 3.1.0 software (FUJIFILM VisualSonics. Toronto, ON, Canada).

### 4.7. Histological Examination

At 8 weeks after the TAC or sham surgery, the mice were weighed and euthanized. The heart was harvested and was flushed with phosphate-buffered saline (PBS) to remove the blood and was then fixed with 4% paraformaldehyde for 72 h, embedded in paraffin, and cut into 5 μm sections. The sections were stained with HE, WGA, and Masson trichrome stain to evaluate the histomorphology, cardiac hypertrophy and fibrosis using a digital scanning imaging system Olympus FV1000 (Olympus, Tokyo, Japan). The myocardial cell cross-sectional area and myocardial fibrosis area were analyzed and calculated using Image J 1.53k software.

### 4.8. Phalloidin Staining

The Phalloidin staining experiments were performed with individual H9c2 cells to study the cell morphology. Briefly, the H9c2 cells were washed with PBS twice and then fixed with 4% paraformaldehyde at 4 °C for 15 min. Next, 0.2% Triton X-100 was applied for permeabilization for 5 min at room temperature. The cells were then washed with PBS three times and were incubated with Phalloidin at room temperature for 20 min. After washing with PBS three times again, the cell sections were observed under a confocal laser scanning microscope (ZEISS, Oberkochen, Germany, LSM900). Image pro plus software was used to quantify the area of the cells.

### 4.9. Western Blotting

As described previously, the mice hearts were harvested and quickly washed with PBS to remove the blood, and protein samples from the myocardium and cells were extracted. The heart tissue was cut into 2 mm thickness and then the protein was extracted from the left ventricular samples using a RIPA system. Then protein concentration was determined by bicinchoninic acid assay (Solarbio co, LTD, Shanghai, China). Extracted protein was separated by 8–12% SDS-PAGE and transferred onto a polyvinylidene fluoride membrane (Millipore, MA, USA). After being blocked and dissolved, the membranes were incubated with the primary antibodies against LC3II, P62, mTOR, p-mTOR, BAX, BCL-2, p-Akt, Akt, and GAPDH overnight at 4 °C. The protein bands were detected by a chemiluminescent system, and the bands were scanned and quantified by densitometric analysis using Image Lab 3.0.

### 4.10. Tandem mRFP-GFP-LC3 Fluorescence Microscopy

The autophagy in H9c2 cells was analyzed using tandem mRFP-GFP-LC3 fluorescence microscopy. Briefly, the H9c2 cells were seeded on the confocal plate until the confluence rate reached about 50%. The cells were transfected with adenovirus expressing mRFP-GFP-LC3 (hanbio, Shanghai, China) at 20 MOI for 2 h. After transfection, the cells were treated as above. The cells were then observed under a confocal laser scanning microscope (ZEISS, LSM900). Autophagosomes (APs) and autolysosomes (ALs) were identified by yellow dots and red dots in merged images. The number of APs and Als, which represent the level of autophagic flow, were calculated and analyzed using Image J software.

### 4.11. Statistical Analysis

All values are shown as mean ± SEM. Normal distributions were verified by the Shapiro–Wilk test. The statistical significance among multiple groups was compared by ANOVA and followed by Bonferroni’s test for multiple comparisons. All statistical analyses were performed using GraphPad Prism software (version 8.0, USA). *p* < 0.05 was considered statistically significant.

## Figures and Tables

**Figure 1 ijms-24-00904-f001:**
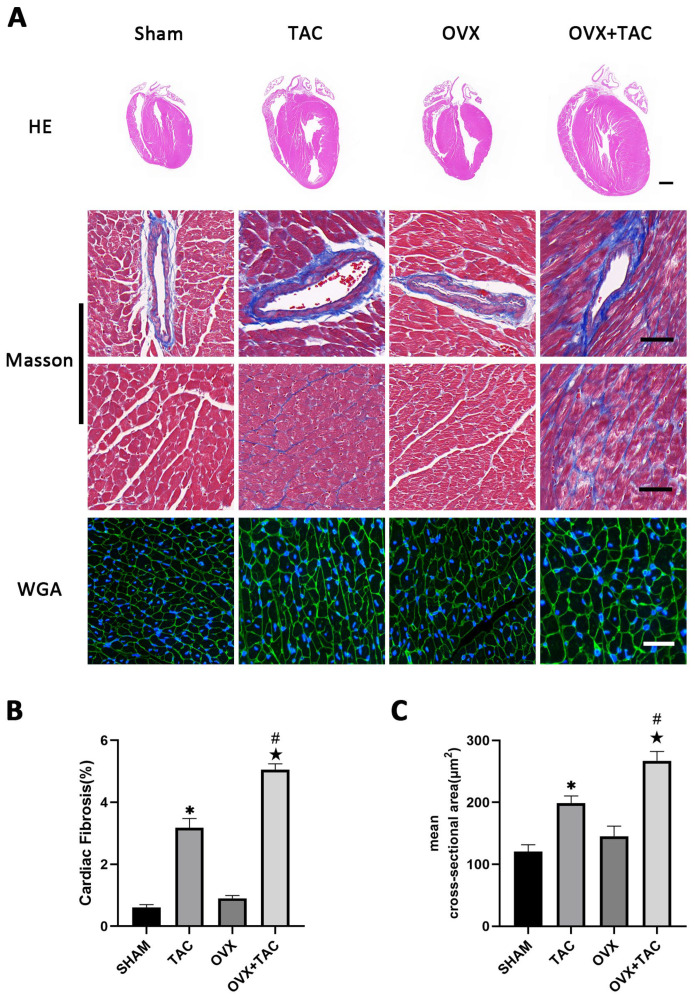
Effects of pressure overload on cardiac hypertrophy and fibrosis in Sham and OVX mice. (**A**) Representative images of cardiac sections for HE (scale bar = 1 mm), Masson and WGA staining (scale bar = 50 µm). (**B**) Cardiac fibrosis ratio in different groups. (**C**) Mean cross-sectional area of cardiomyocytes in each group. The data is expressed as mean ± SEM and was analyzed by one-way ANOVA (n = 4). * *p* < 0.05 vs. Sham; # *p* < 0.05 vs. transverse aortic constriction (TAC); ★ *p* < 0.05 vs. OVX.

**Figure 2 ijms-24-00904-f002:**
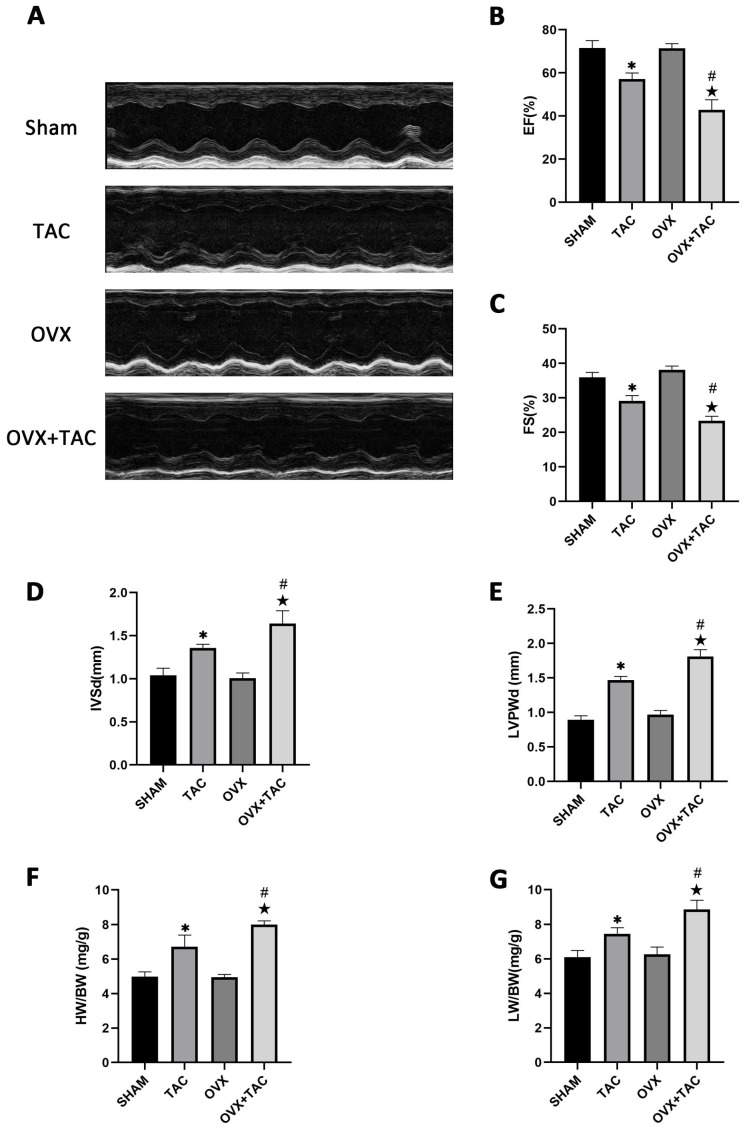
Effect of pressure overload on cardiac function, heart weight/body weight (HW/BW) and lung weight/body weight (LW/BW) after 8 weeks of TAC by M-mode echocardiography. (**A**) Representative images of M-mode echocardiography in different groups. (**B**) Ejection fraction (EF). (**C**) Fraction shortening (FS). (**D**) Interventricular septal thickness at end diastole (IVSd). (**E**) Left ventricular posterior wall thickness at end diastole (LVPWd). (**F**) HB/BW. (**G**) LW/BW. The data is expressed as mean ± SEM and analyzed by one-way ANOVA (n = 4). * *p* < 0.05 vs. Sham; # *p* < 0.05 vs. TAC; ★ *p* < 0.05 vs. OVX.

**Figure 3 ijms-24-00904-f003:**
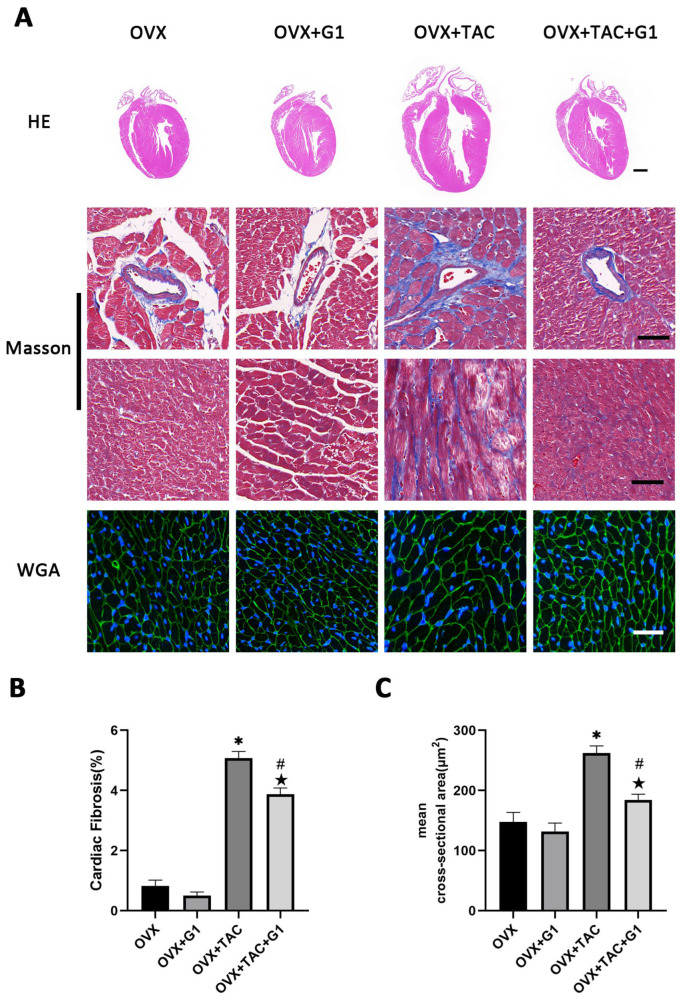
G1 alleviated cardiac hypertrophy and fibrosis of mice in the OVX + TAC group. (**A**) Representative images of cardiac sections for HE (scale bar = 1 mm), Masson and WGA staining (scale bar = 50 µm) in four groups. (**B**) Cardiac fibrosis ratio in different groups. (**C**) Mean cross-sectional area of cardiomyocytes in each group. The data is expressed as mean ± SEM and analyzed by one-way ANOVA (n = 4). * *p* < 0.05 vs. Sham; # *p* < 0.05 vs. TAC; ★ *p* < 0.05 vs. OVX.

**Figure 4 ijms-24-00904-f004:**
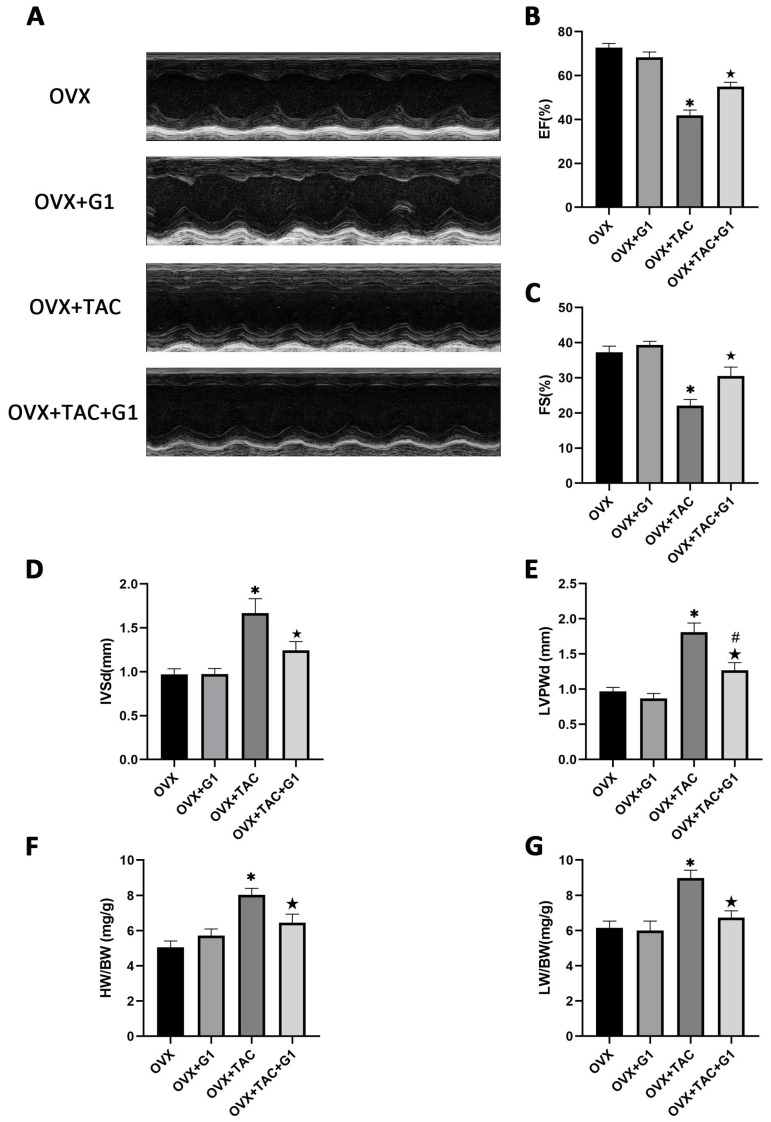
G1 improved cardiac function and attenuated cardiac hypertrophy in OVX + TAC mice by M-mode echocardiography. (**A**) Representative images of M-mode echocardiography in different groups. (**B**) EF. (**C**) FS. (**D**) IVSd. (**E**) LVPWd. (**F**) HB/BW. (**G**) LW/BW. The data is expressed as mean ± SEM and was analyzed by one-way ANOVA (n = 4). * *p* < 0.05 vs. OVX; # *p* < 0.05 vs. OVX + TAC; ★ *p* < 0.05 vs. OVX + G1.

**Figure 5 ijms-24-00904-f005:**
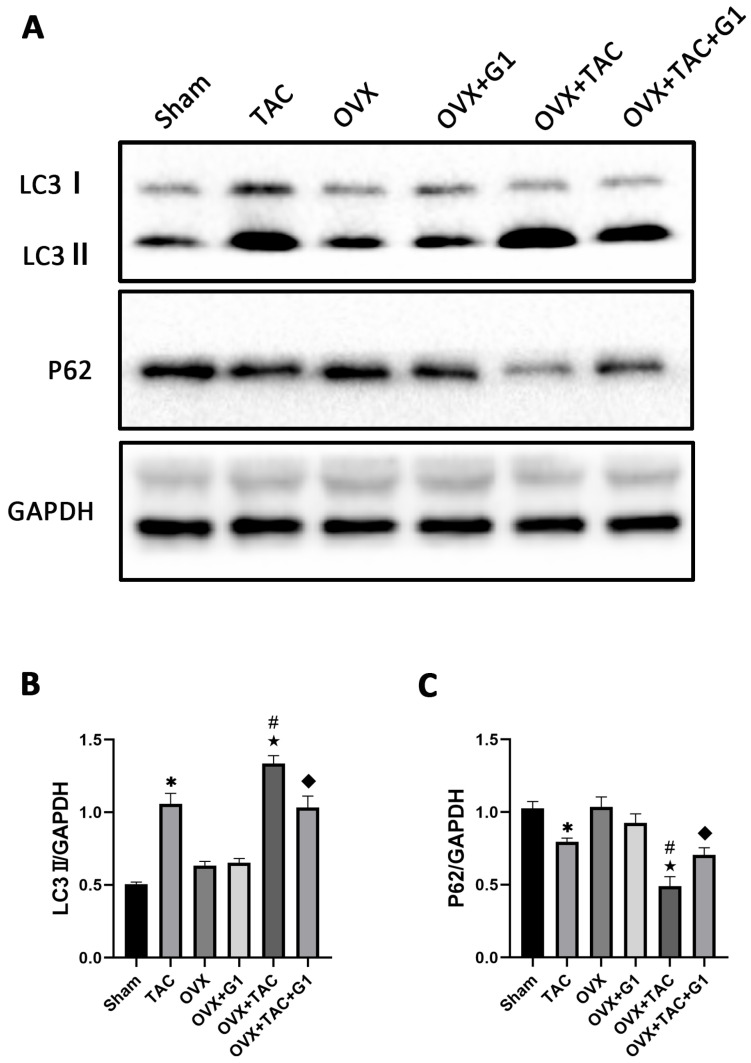
G1 decreased the autophagy of myocardial tissue in OVX + TAC mice. (**A**) Representative images of Western blot in Sham, TAC, OVX, OVX + G1, OVX + TAC, and OVX + TAC + G1 groups. (**B**) Expression analysis of LC3II among different groups. (**C**) Expression analysis of P62 among different groups. The data is expressed as mean ± SEM and was analyzed by one-way ANOVA (n = 3). * *p* < 0.05 vs. Sham; # *p* < 0.05 vs. TAC; ★ *p* < 0.05 vs. OVX; ◆ *p* < 0.05 vs. OVX + TAC.

**Figure 6 ijms-24-00904-f006:**
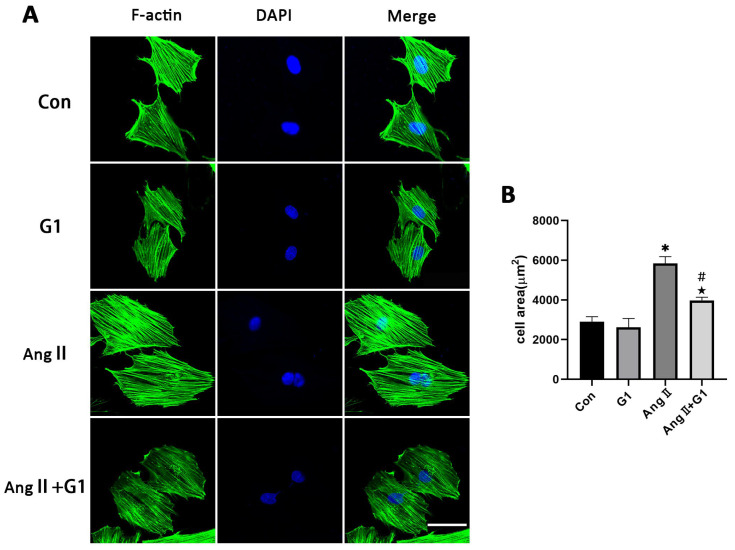
G1 attenuated AngII-induced hypertrophy of H9c2 cells in vitro. (**A**) Representative confocal microscopy images of the H9C2 cells by Phalloidin Staining (scale bar = 50 µm). (**B**) cell area of H9c2s from different groups. The data is expressed as mean ± SEM and was analyzed by one-way ANOVA (n = 4). * *p* < 0.05 vs. CON; # *p* < 0.05 vs. G1; ★ *p* < 0.05 vs. AngII.

**Figure 7 ijms-24-00904-f007:**
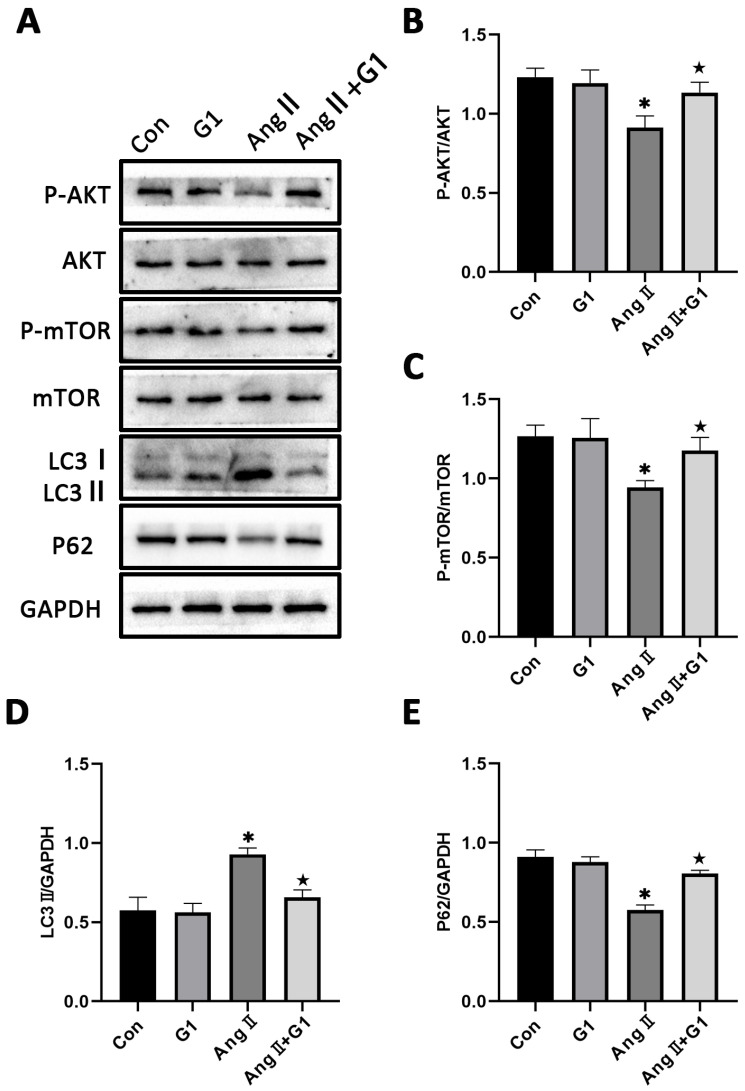
The Akt/mTOR pathway and autophagy were involved in G1 alleviating AngII-induced hypertrophy of H9c2 cells. (**A**) Representative images of Western blots in the Con, G1, AngII and AngII + G1 groups. (**B**) p-AKT/AKT ratios, (**C**) p-mTOR/mTOR ratios, (**D**) LC3II/GAPDH, (**E**) P62/GAPDH from indicated groups. The data is expressed as mean ± SEM and was analyzed by one-way ANOVA (n = 3). * *p* < 0.05 vs. Con; ★ *p* < 0.05 vs. AngII.

**Figure 8 ijms-24-00904-f008:**
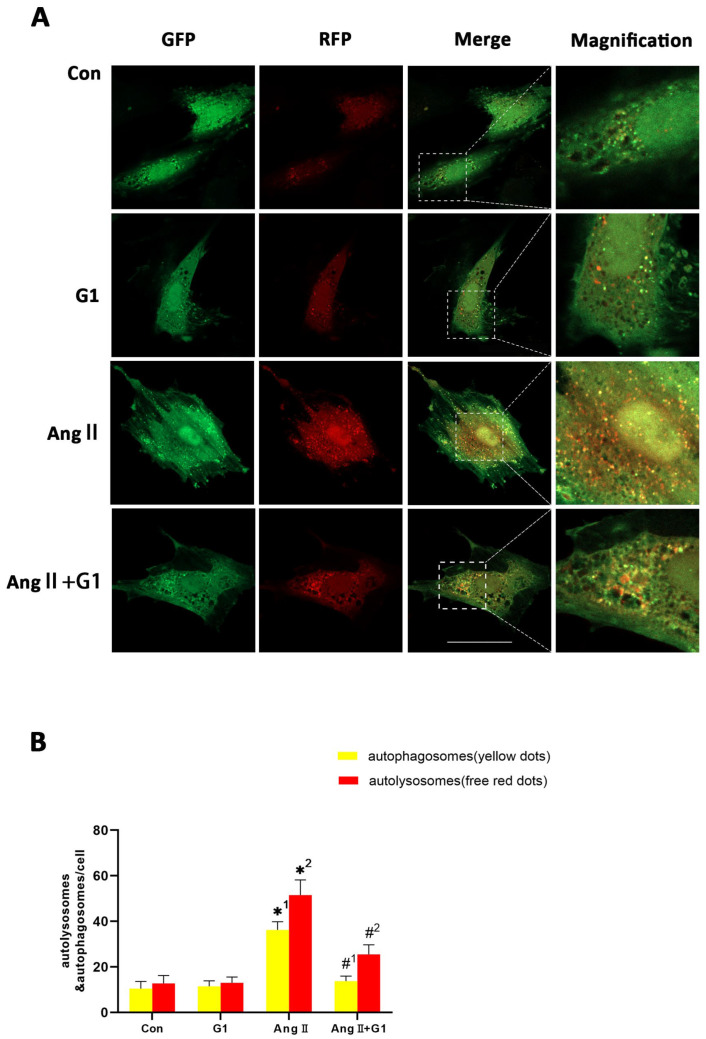
G1 decreased the autophagic flux of H9c2 cells enhanced by AngII. (**A**) Representative confocal microscopy images of LC3 puncta, GFP dots, mRFP dots, and their merged images. (**B**) Mean numbers of autophagosomes (yellow dots) and autolysosomes (free red dots) per cell. The data is expressed as mean ± SEM and was analyzed by one-way ANOVA (n = 4). *1: versus Con group (yellow/APs dots/cell); *2: versus Con group (free red/ALs dots/cell); #1: versus AngII group (yellow/APs dots/cell); #2: versus AngII group (free red/ALs dots/cell). Scale bar = 50 μm.

**Figure 9 ijms-24-00904-f009:**
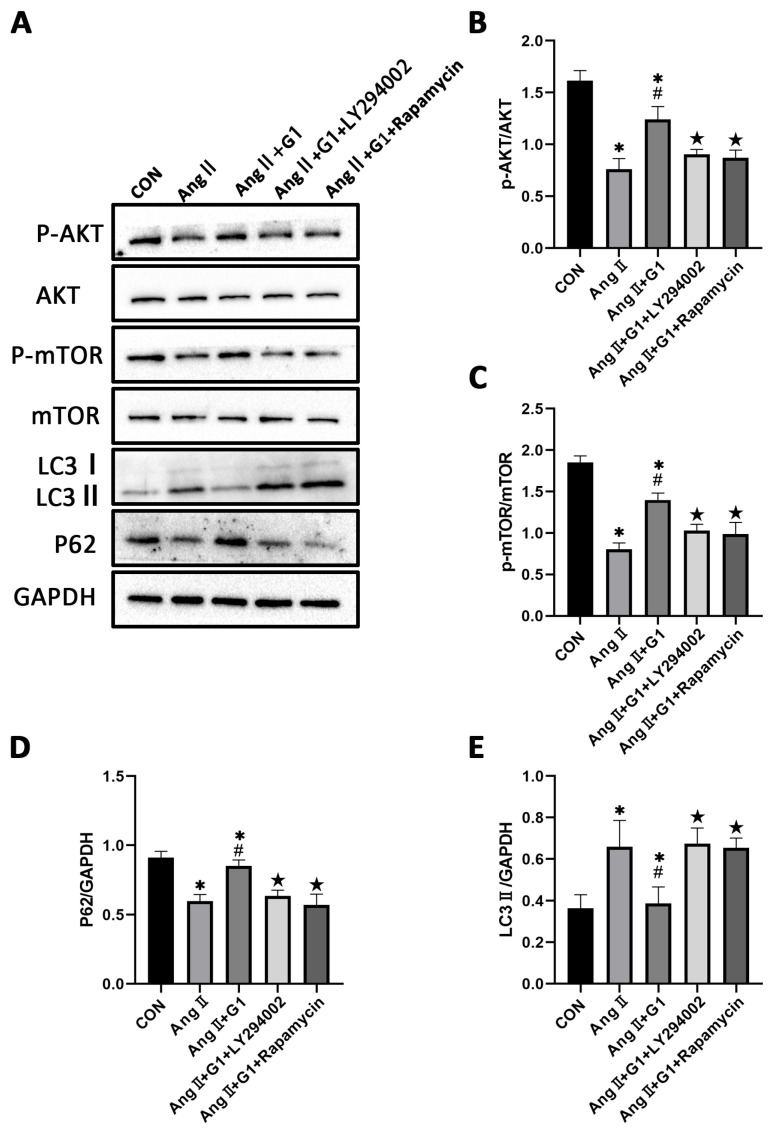
LY294002 or rapamycin blocks the effect of G1 on autophagy induced by AngII in H9c2 cells. (**A**) Representative images of Western blots in Con, AngII, AngII + G1 and AngII + G1 + LY294002, and AngII + G1 + Rapamycin groups. (**B**) p-AKT/AKT ratio, (**C**) p-mTOR/mTOR ratio, (**D**) LC3II/GAPDH, and (**E**) P62/GAPDH from indicated groups. The data is expressed as mean ± SEM and was analyzed by one-way ANOVA (n = 3). * *p* < 0.05 vs. Con; # *p* < 0.05 vs. AngII; ★ *p* < 0.05 vs. AngII + G1.

**Figure 10 ijms-24-00904-f010:**
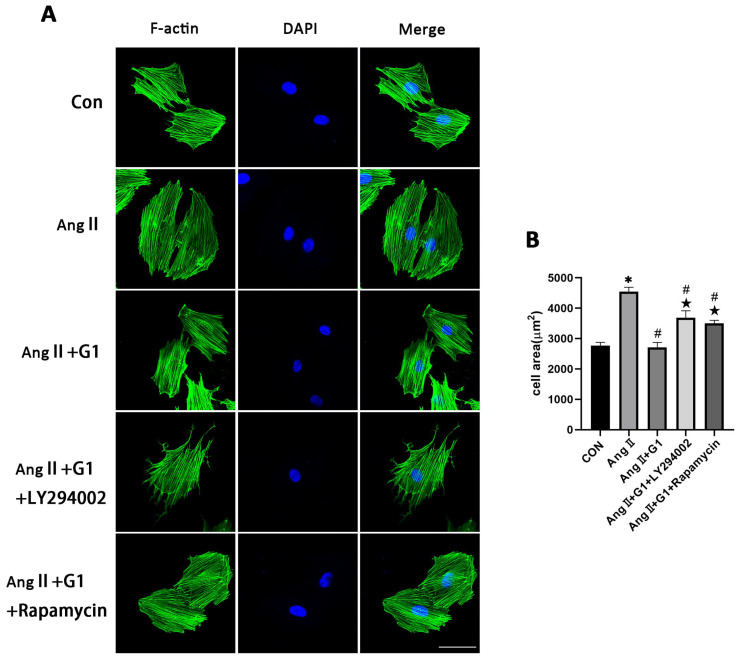
Effect of LY294002 or rapamycin on cardiac hypertrophy of H9c2 cells in the AngII + G1 group. (**A**) Representative confocal microscopy images of the H9C2 cells by Phalloidin Staining (scale bar = 50 µm). (**B**) cell area of H9c2s from indicated groups. The data is expressed as mean ± SEM and was analyzed by one-way ANOVA (n = 4). * *p* < 0.05 vs. Con; # *p* < 0.05 vs. AngII; ★ *p* < 0.05 vs. AngII + G1.

## Data Availability

Data is contained within the article and Appendix A.

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
