# Peer review of "GPR30 Alleviates Pressure Overload-Induced Myocardial Hypertrophy in Ovariectomized Mice by Regulating Autophagy"

_ijms, 2023, doi:10.3390/ijms24020904_

Round 1

Reviewer 1 Report

In the manuscript by Zhang et al., the authors present a well-written study. They showed that ovariectomized mice encounter more severe left ventricular hypertrophy and myocardial fibrosis then sham mice after induction of pressure overload, and that G1 could attenuate these effects. This indicates that estrogen might play a protective role in TAC-induced cardiac hypertrophy and fibrosis. Furthermore, they showed that expression levels of autophagy markers were increased in OVX+TAC mice compared to TAC group and that G1 could reduce these levels. In addition, they used H9c2 cells treated with Ang-II to mimic pressure overload in vitro. Cells treated with Ang-II showed increased hypertrophy and increased autophagic flux , which could be alleviated by G1. Within this in vitro model, Western blot results showed that Akt/mTOR pathway was inhibited in Ang-II treated cells, while it was restored in Ang-II + G1 treated cells. To further verify the mechanism behind G1 alleviated cardiac hypertrophy and autophagic flux, cells were additionally treated with the PI3K inhibitor LY294002 and the autophagy activator rapamycin. There results show that LY294002 and rapamycin could block the   antihypertrophic effect of G1 in H9c2 cells.

The authors claim that these findings provide new insights for potential new choice for drug treatment of pressure overload-induced cardiac hypertrophy in postmenopausal woman. Although a very nice study design, time consuming experiments, interesting results and well written manuscript, a few critical remarks have to made:

1.      In line 273-274 the authors claim that their results showed that GPR30 activation protected TAC induced cardiac hypertrophy in OVX mice by regulating autophagy through AKT/mTOR pathway. And in line 352-353 they claim that GPR30 activation by G1 could attenuate TAC-induced cardiac hypertrophy and fibrosis in OVX mice through regulating autophagy of myocardial tissue by Akt/mTOR pathway. However, currently described results are too weak to make these statements. The authors only measured Akt/mTOR protein levels in the H9c2 cell line, but they did not measure any protein levels of the Pi3k/Akt/mTOR pathway in the in vivo mouse model. Western blot data (protein levels of proteins involvement in Pi3k/Akt/MTOR pathway) have to be provided for SHAM, TAC, OVX, OVX+G1,OVX+TAC, and OVX+TAC+G1 mice.

2.      In addition, it would be interesting to explore involvement of other pathways activated by the G-protein coupled receptor (such as PKA, ERK1/2, HDAC5) as well. Perhaps there might be a synergistic effect of multiple pathways.

3.      In vitro and ex vivo models are essential to investigate mechanisms of disease development and progression and provide a good base for exploring potential new drug targets. But why did the authors choose H9c2 cells for their in vitro model? The healthy adult heart primarily relies on fatty acid metabolism for energy production. Therefore caution should be taken when studying metabolic pathways (such as Pi3k/Akt/mTOR pathway or Autophagy pathway) in glycolytic cell models (H9c2, nRCM and HL1 cell models) as these display significantly lower fatty acid metabolism compared to the healthy adult heart. The H9c2 cell line might not be a representative model. Ref.(M. Rech et al., Assessing fatty acid oxidation flux in rodent cardiomyocyte models, Scientific Reports 2015.)

4.      The authors mention that multiple observation time points might show dynamic changes in myocardial hypertrophy and autophagy. I agree completely on this point, since it could provide additional information with respect to the development of cardiac hypertrophy and it would allow to link autophagic flux to structural remodelling of the heart during the development of heart failure. In addition, it would be interesting to translate findings of the TAC model (acute model of PO-induced heart failure) to a less acute model of PO-induced heart failure (for instance Spontaneously Hypertensive Rats or mice with a Subcutaneous Angiotensin II Infusion using Osmotic Pumps). Since heart failure in post-menopausal woman will develop slowly over time, confirming these findings in a less acute model of PO-induced heart failure would strengthen the findings of this current study even further. Such a study, with multiple observation time points, would also provide the opportunity to determine whether early administration of G1 in post-menopausal woman could prevent cardiac remodelling.

Reviewer 2 Report

GPR30 alleviates pressure overload-induced myocardial hyper-2 trophy in ovariectomized mice by regulating autophagy

In the present work the authors reported a potential mechanism underlying the high percentage of development of heart failure in aged female if compared to men of the same age. Indeed, reduction of sexual hormones in post-menopausal women may represent the cause of the increase of cardiac hypertrophy and fibrosis.

The authors found that activation ofnovel estrogen 12 receptor, G Protein Coupled Receptor 30(GPR30), could attenuate cardiac hypertrophy and fibrosis, improve cardiac function of mice in OVX+TAC group through AKT/mTOR mediated inhibition of autophagy.

Overall the work is well written and results are well presented, and it it suitable for publication in the journal after minor revision

Major

-       In vitro model is represetented by H9C2 cells, which are cardiomyoblast and not completely suitable for hypertrophy study due to the high proliferation rate. Moreover, in materials and methods it is not clear if the authors used a starvation before applying the stimuli. Please add informations. Additionally, hypertrophy activation must be corroborated using specific biomarker (NPPB/NPPA ACTA MYH6 etc). I may suggest to provide a gene expression analysis in all conditions, in order to reinforce data derived from area measurement.

-       In figure 8 I may suggest to put a magnification of the colocalized dots for authopahgosomes/lysosomes.

Minor

-       In the abstract the authors referred to G1 molecules, however it is not reported the function. In order to put the reader in a better condition for understanding the hypothesis and the underlying mechanism, please provide one sentence describing the role of. G1 in modulation of GPR30 signaling.

-       The author should check among the test, some mistakes like no space between “group(FIGURE….) line 192. This mistake is present in different points along the test. Please correct
